# Fine-Mapping of Sorghum Stay-Green QTL on Chromosome10 Revealed Genes Associated with Delayed Senescence

**DOI:** 10.3390/genes11091026

**Published:** 2020-09-01

**Authors:** K. N. S. Usha Kiranmayee, C. Tom Hash, S. Sivasubramani, P. Ramu, Bhanu Prakash Amindala, Abhishek Rathore, P. B. Kavi Kishor, Rajeev Gupta, Santosh P. Deshpande

**Affiliations:** 1International Crops Research Institute for the Semi-Arid Tropics, Patancheru, Hyderabad 502324, India; knskira@gmail.com (K.N.S.U.K.); s.sivasubramani@cgiar.org (S.S.); punnaramu@gmail.com (P.R.); bprakash.a@gmail.com (B.P.A.); a.rathore@cgiar.org (A.R.); g.rajeev@cgiar.org (R.G.); 2Department of Genetics, Faculty of Sciences, Osmania University, Hyderabad, Telangana 500007, India; 3International Crop Research Institute for the Semi-Arid Tropics, Niamey BP12404, Niger; ct_hash@yahoo.com; 4Department of Biotechnology, Vignan’s Foundation for Science, Technology & Research, Vadlamudi, Guntur, Andhra Pradesh 522213, India; pbkavi@yahoo.com

**Keywords:** sorghum chromosome 10 long arm (SBI-10L), fine-mapping, stay-green, candidate genes, percent green leaf area, delayed senescence

## Abstract

This study was conducted to dissect the genetic basis and to explore the candidate genes underlying one of the important genomic regions on an SBI-10 long arm (L), governing the complex stay-green trait contributing to post-flowering drought-tolerance in sorghum. A fine-mapping population was developed from an introgression line cross—RSG04008-6 (stay-green) × J2614-11 (moderately senescent). The fine-mapping population with 1894 F_2_ was genotyped with eight SSRs and a set of 152 recombinants was identified, advanced to the F_4_ generation, field evaluated with three replications over 2 seasons, and genotyped with the GBS approach. A high-resolution linkage map was developed for SBI-10L using 260 genotyping by sequencing—Single Nucleotide Polymorphism (GBS–SNPs). Using the best linear unpredicted means (BLUPs) of the percent green leaf area (%GL) traits and the GBS-based SNPs, we identified seven quantitative trait loci (QTL) clusters and single gene, mostly involved in drought-tolerance, for each QTL cluster, viz., AP2/ERF transcription factor family (*Sobic.010G202700*), NBS-LRR protein (*Sobic.010G205600*), ankyrin-repeat protein (*Sobic.010G205800*), senescence-associated protein (*Sobic.010G270300*), WD40 (*Sobic.010G205900*), CPK1 adapter protein (*Sobic.010G264400*), LEA2 protein (*Sobic.010G259200*) and an expressed protein (*Sobic.010G201100*). The target genomic region was thus delimited from 15 Mb to 8 genes co-localized with QTL clusters, and validated using quantitative real-time (qRT)–PCR.

## 1. Introduction

Sorghum is the fifth most important cereal crop with a thick, waxy cuticle known to be better adapted to arid, semi-arid tropical, and sub-tropical climatic conditions, serving as a staple food for many of the world’s poorest and food-insecure people [1]. On an average, sorghum is grown in 40.67 million ha globally, accounting for an annual production of 57.60 million tons during 2017 (http://www.fao.org/faostat). A plethora of biotic and abiotic stresses are the major constraints of sorghum, while drought is a major abiotic constraint behind the significant loss in crop productivity across the world. Delay in senescence is one of several mechanisms that can contribute to the ability of a plant to withstand drought stress [2]. Such functional “stay-green” (stg) individuals retain green leaf area (GL) for a longer period of time, following the onset of a “drought spell”, and this can be expected to have a more stable grain yield performance across sites and years, in their zones of adaptation. The best-characterized trait contributing to grain-yield maintenance under terminal drought stress/post-flowering drought tolerance is “stay-green”, which is well-documented in several economically important crop plants like sorghum [3], maize [4], wheat [5], barley [6], rice [7], and Arabidopsis [8].

Studies in the model plant Arabidopsis explained several mechanisms involved in (leaf) senescence—autophagy, chlorophyll catabolism, and nutrient remobilization [8]. Previous studies indicated that several gene complexes [9,10], phytohormones [11], and environmental factors [12] are involved in the senescence mechanism. Moreover, alteration of any of the factors might result in a stay-green phenotype. Other ways to stay-green include water-saving mechanisms during the pre-reproductive growth stages [13], which might compromise the yield potential (e.g., via a reduced tillering, or a reduced flag leaf area)—but could still be economically acceptable in specific crop production environments. The risk of drought stress is sufficiently high to make a selection for maximum grain-yield potential, and it is too risky as a breeding target.

Stay-green is the best-characterized component of drought tolerance and enhances the yield of sorghum grain under drought, by modifying canopy development, root growth, and crop-water usage. The presence of stay-green prevents stalk lodging and charcoal rot, while maintaining normal grain filling [13,14,15]. Several Quantitative Trait Loci (QTL) mapping studies that assess stay-green expression under drought stress conditions were undertaken using mapping populations [3,16], introgressed lines [17], and near-isogenic lines [18]. Leaf senescence is a complex process that involves thousands of genes that are upregulated and downregulated, based on various developmental stages, and environmental, physiological, and chemical factors [9,10,19]. Among different, stay-green sources identified, B35 is familiar and has been exploited by many research groups, as compared to E36-1. Many studies on the E36-1 stay-green population identified major stay-green QTLs on SBI-10, and these QTLs are overlapped with root angle QTLs [3,20]. The large stg QTL interval on SBI-10L is a barrier to identify QTL/genes for marker-assisted selection (MAS). To narrow down stg QTL, genotyping by sequencing (GBS) has increased thousands of SNP markers, which allows us to fine-map the targeted QTLs. Fine-mapping of stg QTLs using GBS is a powerful method to identify the causal genes responsible for trait expression. Given the diversity of stay-green mechanisms, clarity is required about how the trait is regulated/controlled; thus, further defining the genetic basis of stay-green for efficient deployment of useful alleles through marker-assisted breeding. To achieve this, a sub-set of 152 F_2:4_ progenies were generated. These had recombination events in the target region of SBI-10L and were utilized in this fine-mapping study. We further report the dissection of the major stay-green QTLs in the target genomic region into several components. Using GBS markers and the putative candidate genes, the underlying individual components of this QTL were validated using the quantitative real-time (qRT) PCR. This study paved the way for more detailed fine-mapping of this target region under drought-stress, along with an improved grain yield. Additionally, it provided an opportunity for cloning these candidates that contribute to the stay-green phenotype, while simultaneously generating economically desirable recombinants that are expected to be particularly useful in the drylands of South Asia, especially peninsular India and sub-Saharan Africa.

## 2. Materials and Methods

### 2.1. Parents and Fine-Mapping Population Development

Stay-green fine-mapping population of 1894 F_2_ was developed by crossing an introgression line cross RSG04008-6 (with stay-green QTL on SBI-10L; a BC_2_F_3:4_-derivate from stay-green exotic donor B35, an Ethiopian durra, with recurrent parent R16, a post-rainy Indian durra) × J2614-11 (with shoot fly resistant QTL on SBI-10L; a BC_3_F_3:4_-derivate from shoot fly resistance exotic donor IS18551, an Ethiopian durra, with recurrent parent BTx623, standard B-line). The F_2_ population was initially screened with 5 SSRs, across the target QTL interval, including Xgap001 and Xtxp141 SSRs (Figure 1A). For better coverage and detecting the possible recombination in the complete stay-green QTL region, three additional SSRs were screened across the complete F_2_ population [21]. Based on the recombination events between target SSRs covering the homozygote recombination events for every interval, a sub-set of 152 F_2_ genotypes were selected for population advancement and genotyping (GBS) (Figure 1B). Selected F_2_ recombinants were advanced by the single seed descent method, and F_2:4_ recombinants were evaluated in a replicated field trial (Figure 1). Graphical genotypes in Figure 1 are represented using the GGT 2.0 software [22].

### 2.2. Field Evaluation of Stay-Green

F_2:4_ progenies and parents were evaluated in different environments, i.e., the post-rainy season of 2012–2013 (E1/13) and 2013–2014 (E2/14) at ICRISAT, Patancheru (17°30′ N, 78°16′ E, and altitude 545 m), India. The onset of stress was manipulated by controlling the supply of water before flowering. Irrigation was given only once to the experimental field, after sowing. Seeds were sown on a shallow (40 to 60 cm) vertical inceptisol (very fine montmorillonite isohyperthermic) overlying a loose, decomposing, granite-based material that is permeable to roots, but which contains limited plant-available water. The plant material including 152 F_2:4_ progeny, parents in duplicates (4), were replicated thrice and arranged in an alpha lattice design with 39 blocks per replication × 12 entries per block [23]. The experimental units were 2-row plots (length 2 m) with 45 cm and 15 cm inter- and intra-row spacing, respectively. A basal application of 20 kg ha^−1^ N and 20 kg ha^−1^ P2O5 as di-ammonium phosphate was used before sowing. For E1, sowing was carried out on 31st December 2012, and in E2, on 20th November 2013. Nearly 40-days of sowing differences were observed between E1 and E2. The seeds were machine sown, and the field was irrigated with overhead sprinklers to ensure germination. The crop was thinned 10-days after emergence to about 60,000 plants ha^−1^. The temperature regime was different in the two seasons, due to the different dates of sowing and hence, exposure to stress variations. The mean monthly temperature ranged from 30 to 40 °C and 28 to 37 °C, respectively for E1 and E2. Rainfall of 60 mm and 85 mm was received, respectively, for E1 and E2 (Appendix A).

### 2.3. Estimation of Senescence and Grain Yield

Leaf senescence pattern was assessed plot-wise under partially-irrigated conditions. The %GL of each plot was estimated visually on a weekly basis from anthesis (days to 50% flowering) to harvest. Visually, senescence was recorded during the booting stage, from the bottom leaves to the top booting leaves, by comparing with the adjacent rows. The visual senescence readings were deducted from 100 to get the %GL. The senescence scores ranged from 10% to 100% senescence, and weekly senescent scores were converted to %GL scores (for example %GL = total green leaf area − senescence; 100% − 10% = 90%), which were named as percent green leaf area (7–49) days after flowering (%GL7, %GL14, %GL21, %GL28, %GL35, %GL42, and %GL49 DAF) [24]. Grain yield (GY) was measured with component traits like panicle dry weight in grams/plot (PnDW), grain dry weight in grams/plot (GDW), mean 100-grain mass (g) (HGM), grain number per plot and panicle (GNP and GNPP), and panicle harvest index (PHI).

### 2.4. Statistical Analysis

Analysis of variance (ANOVA) and the phenotypic correlations were performed using the GenStat (14th edition). For the alpha lattice design, the best linear unpredicted means (BLUP) were estimated using the residual maximum likelihood algorithm (ReML) for E1, E2, and across environment [25], to determine the Genotype × Environment (G × E) effect. For each entry and each trait, the predicted means were calculated, with replication as a fixed effect and the blocks within replication and genotype as the random effect (Rep/Block + Geno).

### 2.5. Heritability

It was estimated in Recombinant Inbred Lines (RILs) for all resistance components, as well as the ratio of total genotypic variance to the phenotypic variance [26].
$$H2=\frac{\sigma_{g}^{2}}{\sigma_{p}^{2}}\times 100$$
where, *H*2—% heritability coefficient, *σ_g_*—genotypic variance, *σ_p_*—phenotypic variance; 0–30%—low, 30–60%—moderate, and 60% and above—high heritability percentages.
$$S.E.=\sqrt{\frac{N-1}{N}\frac{\left(\mathrm{Error}\;\mathrm{MS}\right)}{r}}$$
where,
S.E. = Standard ErrorN = Number of individualsError MS = Error mean sum of square.

$$\mathrm{CV}=\frac{\sqrt{\mathrm{Error}\;\mathrm{MS}}}{\mathrm{GM}}\times \;100$$
where,
CV = Coefficient of VariationError MS = Error mean sum of squareGM = Grand mean.

### 2.6. Genotyping by Sequencing (GBS), SNP Calling, and Annotation

The DNA from cross RSG04008-6 × J2614-11, along with their parents (in duplicates), was isolated and the GBS data [27] for a total of 156 (152 + 4) genotypes were generated. Genomic DNA samples were digested individually with ApeKI (recognition site: G|CWCG). The fragments were ligated with sample-specific “barcodes” called “restriction site-associated DNA tags” (RAD tags) and restricted, while the barcoded DNA samples were then multiplexed at 96-plex, in two lanes of an Illumina HiSeq2000. GBS libraries were constructed and subjected to skim sequencing, to a depth of 0.1X. The resulting 66-base pair sequence reads were used. Sequences were mapped to the BTx623 sorghum reference genome V3.1 [28], whereas, newly identified genes were annotated using phytozyme V12 (https://phytozome.jgi.doe.gov/pz/portal.html#!info?alias = Org_Sbicolor). Single Nucleotide Polymorphism (SNP)s were called using the TASSEL v4.3.10 GBS pipeline [29]. Sequence tags, 64-bp sequences that included a leading 4-bp C[T/A]GC signature from the cut site were identified, and tags with at least 10X total coverage read depths with 0.01 minor allele frequency were retained. SnpEff V4.3 [30] was utilized for the functional annotation of SNPs identified using the reference gene feature information. Parents RSG04008-6 was coded as ‘A’, J2614-11 as ‘B’, and heterozygote as ‘H’. The CMplot (https://github.com/YinLiLin/R-CMplot) R package was utilized to visualize the genome-wide density of SNPs.

### 2.7. Distance Matrix and Principal Coordinate Analysis (PCA)

A distance matrix between all selected pairs of markers (1515 SNPs + 8 SSRs [Simple Sequence Repeats] were used to identify the meiotic recombination events]) was calculated using the THREaD Mapper Studio [31] and was used for the principal coordinate analysis (PCA).

### 2.8. SSR-SNP High-Resolution Linkage Map Construction and QTL Analysis

The markers in the linear order of the horseshoe arch were used for linkage map construction, with the help of JoinMap v3.0 [32]. The Kosambi map function was used to convert the recombination fractions into centimorgans (cM) [33]. Marker order was assigned at minimal LOD 3, and segregation distortion, and chi-square values were calculated using the JoinMap v3.0 [32]. QTL mapping for the 152-entry F_2:4_ population was performed using composite interval mapping (CIM), which was implemented in QTL Cartographer Windows v2.5 [34], with default settings (window size of 10cM, walking speed of 1cM, control markers = five, and backward regression). The significance of each QTL interval was determined with the threshold level estimated at 1000 permutations, with *p* ≤ 0.05 for significant QTL detection.

### 2.9. Stg QTL Cluster Analysis and Fine-Mapping

QTLs sharing common marker intervals with common peak positions were grouped into QTL clusters. For fine-mapping F_2_, 152 selected recombinants were aligned with their phenotypic data and sorted with phenotypic values. Both phenotypic extreme haplotypes were aligned to clearly visualize the recombination breakpoint, which helped in the clear identification of the genomic regions. These regions are responsible for the stay-green traits, by avoiding environment and genetic background effects and other unknown factors [35]. The haplotypes, compared for the identification of informative recombinant breakpoints along with phenotyping data, were determined manually.

### 2.10. Marker Trait Associations (MTAs)

As a supportive analysis in the identification of the putative candidate genes for each QTL cluster (as explained in following section), marker-trait associations (MTAs) between the genome-wide SNPs (29,506 SNPs) and stay-green phenotype traits were estimated. Calculated BLUPs for each phenotype of three replicated field data of 152 genotypes were used for the MTA analysis. The R package GAPIT [36] was used to find out significant MTAs of the stay-green QTLs on the sorghum chromosome SBI-10L. We conducted the MLM (Q + K) method in GAPIT to remove false positives in identifying significant marker trait associations. Circos v0.69 (http://circos.ca/) [37] was utilized to visualize significant SNP associations. Total SNPs from the genome and SBI-10 common SNPs were represented in the Venn diagram, using the website-http://bioinformatics.psb.ugent.be/webtools/Venn/.

### 2.11. Candidate Genes Identification

GBS-SNP markers flanking each target QTL (assumed to be under the control of a single gene per QTL) could be combined with the annotated aligned sorghum genome sequence to identify candidate genes that might be associated with each trait. According to earlier studies, candidate genes responsible for stay-green in other plants like maize, *arabidopsis*, and rice were selected based on their functional role. Fine-mapped regions that had candidate genes supported by annotation of MTAs were searched (http://phytozome.jgi.doe.gov/pz/portal.html) and reported as putative candidate genes, based on their physical positions and their functional role.

### 2.12. RNA Extraction, Candidate Gene Primer Designing, and (qRT-PCR) Analysis

Total RNA was extracted from 7-day-old seedling leaf tissue into three biological replicates for each parent, RSG04008-6 and J2614-11, using the Machery-Nagel™ kit. RNA was quantified using the NanoDrop spectrophotometer and converted to cDNA, using the Superscript III cell direct cDNA synthesis kit (Invitrogen, Life technologies, Waltham, MA, USA). The cDNA samples were diluted and normalized with the housekeeping gene actin. coding DNA sequence (CDS) for designing gene-specific primers were extracted from FGENESH, for the candidate genes. A total of 8 candidate gene-based primers were designed using primer3 software with a G/C content above 40% and tm ranging from 58–62 ± °C conditions for expression. The qRT-PCR analysis was performed using the Applied Biosystems 7500 Real-time PCR system with SYBR green chemistry (Applied Biosystems, Waltham, MA, USA), using two technical duplicates for three biological replicates of each parents. The data from different cDNA samples were compared using the mean of the sample cycle threshold (Ct) values of the three biological replicates, which were standardized for each template, using actin as control. The relative expression ratios of each gene were determined using the 2^−∆∆*C*t^ method and the student’s *t*-test was used to calculate the level of significance.

### 2.13. Data Availability Statement

The datasets generated in this study can be found at ICRISAT data repository. ICRISAT has adopted an open source research data repository software—Dataverse (https://dataverse.org). The DOI link for the deposited SNP data in the ICRISAT-repository. https://doi.org/10.21421/D2/TO4JWO and SSR data DOI link: https://doi.org/10.21421/D2/45YOFR.

## 3. Results

### 3.1. Recombinant Selection for Advancement

The stay-green QTLs on SBI-10 with flanking *Xgap001* and *Xtxp141* SSR markers were overlapped with the shoot fly resistance QTLs, between the same flanking SSR markers. For better coverage, additional SSRs viz., Is10253, and Xisep1011 were added to capture complete QTL regions, and these flanking markers were utilized in the present study (Figure 1) [21]. The initial QTL interval reported by Haussmann et al. (2002) was 37 cM [3], which was extended to 72 cM, with additional SSRs mapping as explained above. This increased the ability to detect and map recombination events in the relatively large QTL interval, which in turn resulted in the identification of additional QTL (clusters), indicating the involvement of several gene complexes in the regulation of stay-green expression. The experimental design was carried on NIL parents RSG04008-6, a stay-green donor for the stay-green QTLs on sorghum chromosome SBI-10, and the other parent J2614-11 is a shoot fly resistant QTL donor on SBI-10, which overlaps with the stay-green region [21]. The parents and the population segregated for the overall stay-green expression (including the target region) in this study, focused on the known major QTL on SBI-10L. To identify and characterize the stay-green QTL, this cross was developed, and the fine-mapping population of 1894 F_2_ recombinants was screened for homozygous recombinants in the shoot fly and stay-green overlapping QTL region on the SBI-10L, using Is10263 and Xisep1011 flanking SSRs (Figure 1A,B and Figure 4A,B).

### 3.2. Traits Variation, ANOVA, and Correlation

Best Linear Unbiased Predictions (BLUPs) values of parents RSG04008-6 and J2614-11, and their F_2:4_ fine-mapping population for seven %GL (7–49 Days After Flowering; DAF) observations and six-grain yield-related (PnDW/plot, GDW/plot, PHI, HGM, GNP/plot, GNPP) traits are detailed in Table 1.

The %GL BLUP values for RSG04008-6 were higher than J2614-11 and differed significantly with the *p* value ≤ 0.05 for both E1 and E2 environments, except the %GL recorded at 42 and 49 DAF. Parents and progeny showing a clear variation, with standard error variation ranging from SE ± 5–13 (Appendix A). RSG04008-6, in general, had a higher %GL compared to J2614-11, across seasons (Table 1, Appendix A). The rate of senescence observed between parents and progeny also clearly indicated that RSG04008-6 is a stable stay-green parent as compared to J2614-11. Comparing the mean values of parents in two seasons and average mean values of the F_2:4_ recombinants, the progenies showed a better stay-green performance than their parents (Appendix A). The trait variation was observed in the progeny, and their frequency distributions for individual seasons and across seasons were normal, while few individuals skewed towards the female parent RSG04008-6, and others showed a discontinuous distribution (Appendix A). The ANOVA showed a highly significant (*p* ≤ 0.001) genotype × environment interactions for all traits, except the GNP/plot and the GNPP. Broad-sense heritability (h^2^) values of stay-green traits were moderate to high, which ranged from 52.67 to 81.80% (Table 2). Heritability estimates were high for both the environments, which could be preferred for breeding. G×E values were highly significant (*p* ≤ 0.001), indicating an environmental effect on the stay-green trait (Table 2).

Correlation coefficients for weekly stay-green scores were significant and exhibited a positive correlation with each other and across seasons, but more in the magnitude for E2 as compared to E1. This could be attributed to the difference in stress exposure/regime in the two seasons (Table 3, Appendix A). During E1, the crop attained a flowering stage in March 2013 and experienced severe stress due to heat and drought as compared to E2. Therefore, there could be high transpiration and imposed stress during grain filling, as could be seen from environmental data (Appendix A and Appendix A). Percent of GL7 had a positive correlation with all the grain-yield-related traits, except HGM and PHI. Percent GL14 was positively correlated with grain yield-related traits, except for the GDW/plot_12, PHI_12, PnDw_13, and HGM (both seasons). Percent GL21 and %GL28 were positively correlated to GY traits, except for GDW, GDW/plot, HGM, PHI, PnDw/plot (for both seasons), and PnDw/13. Percent GL35, %GL42 and %GL49 were positively correlated with GY, except for GDW/12, GDW/plot_12, PHI_12, PnDw/12, PnDw/plot_12, and HGM (for both seasons; Table 3).

### 3.3. Genotyping by Sequencing (GBS) and SNP Annotation

We obtained a total of 683,645,045 effective sequence reads covering the whole genome, and 110,768 variants were identified from those reads. After using stringent filters (0.01 MAF) in the TASSEL GBS pipeline, a set of 29,506 SNPs were identified across the sorghum genome from 152 F2 selected recombinants (Appendix A). SNP had transitions (A/G or C/T) of 49% and transversions (A/C, A/T, C/G, or G/T) of 42%. The observed transition–transversion ratio was 1.15 in the raw SNP variants. The highest number of SNPs identified on SBI-01 were 5121 and the lowest number of SNPs on SBI-08 were 1928. The density of SNPs for the 1 Mb region ranged from 32–288 and is represented as a heatmap in Figure 2.

We focused on the recombination events of the SBI-10L region of 45–60 Mb, where 1515 SNPs were identified in the target region (~1SNP/kb), using the SNP effect. We identified non-synonymous (nucleotide substitution results in amino acid change) vs. synonymous (no change in amino acid after nucleotide substitution) SNPs (nearly 24.16%). These SNPs were located in the exonic regions (18.5% synonymous coding regions + 5.6% non-synonymous coding regions), intergenic, and intronic regions (47.72% and 11.32%, respectively) (Appendix A).

### 3.4. Development of High-Resolution Genetic Linkage Map by Integrating GBS–SNP into SSR Map of the SBI-10L

An integrated linkage map with 8 SSR markers for the target flanking region (with 72cM interval with SSRs) was constructed with 152 selfed F_2:4_ individuals, with mostly homozygous step introgressions [21] covering the target region. Allele calls for 8 SSRs and 1515 GBS–SNPs were compared across the parents (RSG04008-6 and J2614-11) and grandparents (E36-1, R16, IS18551, and BTx623) of the population.

A set of 618 SNPs (out of 1515 SNPs) was shortlisted, confirming the allele calls between progenitors (parents and grandparents) and population, from a further analysis involving F2 data. SSR allelic proportions were compared with SNP allelic proportions for allele ‘B’ (Appendix A). These 618 SNPs, along with 8 SSRs, were checked for genotype/allele call duplicates (for a given step introgression) for the SNPs next to each other, indicating redundancy in representation and thus a total of 232 SNP duplicates were removed. Principal Coordinate Analysis (PCA) of the filtered markers produced an arch effect named the Horseshoe effect (Appendix A). The markers lying on the horseshoe line were aligned in the order of markers of the linkage group. Those marker loci placed centrally showed many missing data points and were eliminated from the genetic mapping. Finally, 265 (including 8 SSRs) tightly linked markers were utilized for genetic linkage map construction. A distance matrix was plotted for these 265 markers (Appendix A). The linkage map was constructed for these 265 markers at LOD 3 using a single group, with a total distance of 139.7 cM, using the ‘Kosambi’ mapping function on SBI-10L (Figure 3) (Appendix A). The genetic map order had a good correlation with the physical map order (r^2^ = 0.94) (Appendix A).

### 3.5. Stay-Green QTL Mapping and QTL Co-Localization

Stay-green QTLs were detected for E1/13, E2/14, as well as across environments, using an SSR-SNP high-density map of SBI-10L. A total of 33 QTLs (11, 16, and 6, respectively, for E1, E2, and across) were identified (Table 4 and Appendix A) at seven different weekly scores of stay-green (%GL7, %GL14, %GL21, %GL28, %GL35, %GL42, and %GL49). Four QTLs for %GL7, eight for %GL14, seven for %GL21, four for %GL28, one QTL %GL35, six for %GL42, and three for %GL49 with combined phenotypic variance (PVE) of 32.02%, 53.68%, 49.08%, 15.62%, 2.03%, 22.18% and 8.71%, respectively, were explained.

Interestingly, the additive effect contributed by J2614-11 was also observed for many QTLs that displayed stay-green alleles. The positive additive effect for stay-green QTLs revealed that the alleles were derived from RSG04008-6. Four major QTLs viz., Q10GL7b_14, Q10GL14e_14, Q10GL14a_across, Q10GL21a_across explaining 9.43, 9.70, 10.20, 10.14% PVE, respectively, were identified. Co-localized QTLs were clustered (QTLs falling in defined similar marker intervals) to simplify the fine-mapping analysis. Each co-localized region of QTL was named as “cluster QTL stay-green on SBI-10 (cQstg10)” along with a serial number. From 33 stg QTLs detected, 19 QTLs were clustered into seven groups on SBI-10L (Table 5 and Figure 4).

### 3.6. Fine-Mapping of the Stay-Green QTL Clusters

Identified QTL cluster regions of F_2:4_ recombinants were arranged for haplotype analysis, along with phenotype BLUPs. The genotypic information of recombinants from GBS–SNPs was compared with the high and low stay-green phenotype values, to identify the variant genomic regions responsible for phenotypic variation (Figure 4C). The significant GBS-based SNPs present at the peak position were examined for the SNP variation and delimited to a single gene for each QTL cluster (Table 5 and Figure 4).

The fine-mapped cluster cQTLstg10.1 overlapped an AP2/ERF transcription factor family gene (SNP: S10_52969717; Gene Id: *Sobic.010G202700*), cQTLstg10.2 contained a candidate encoding uncharacterized protein (SNP: S10_54388058; Gene Id: *Sobic.010G201100*), cQTLstg10.3 contained a candidate encoding ankyrin-repeat protein (ARP) (SNP: S10_54891180; Gene Id: *Sobic.010G205800*) and a WD40 repeat protein gene (SNP: S10_54899228; Gene Id: *Sobic.010G205900*), cQTLstg10.4 overlapped a candidate encoding NBS-LRR protein (SNP:S10_54388058; Gene Id: *Sobic.010G201100*), cQTLstg10.5 contained a LEA2 protein (SNP: S10_59620328; Gene Id: *Sobic.010G259200*), cQTLstg10.6 contained a calcium/calmodulin-dependent protein kinase gene (SNP:S10_60059042; Gene Id: *Sobic.010G264400*), 7.5 kb upstream to the SNP S10_59850910 region, and cQTLstg10.7 contained a senescence-associated protein gene (SNP: S10_ 60468746; Gene Id: *Sobic.010G270300*). The first four stg QTL clusters (cQTL10.1, cQTL10.2, cQTL10.3, and cQTL10.4) were encompassed within a 503 kb (54388058bp- 54891180bp) region with 50 annotated genes, and the remaining three QTL clusters (cQTL10.5, cQTL10.6, and cQTL10.7) fell within the 848 kb (59620328bp-60468750bp) region with 109 annotated genes, with 11 and 8 component QTLs respectively. Out of 159 genes (50 + 109) in the fine-mapped region, based on the clusters and their recombination events, only eight candidate genes were short-listed. All reported SNPs showed variation in high and low phenotype values for the individual and across season environments, for their corresponding genotype. Altogether, stay-green fine-mapping remains a complex process, but from our data sets, we inferred that transcriptional factors play crucial roles in post-flowering drought stress. This was made possible with the approach of mapping and identification of overlapping QTL (cluster), with a saturated genetic map with MTA providing supporting analysis.

### 3.7. MTAs for Stay-Green Traits

MTAs for the stay-green trait (overall 7 stay-green scores) were performed in the present bi-parental population, specifically targeting the long arm of SBI-10 in sorghum, as the homozygous genotypes dropped based on eight SSR markers and with the retention of only a diverse recombinant genotype. A total of 29,506 SNPs across the sorghum genome were called. Out of this, 6491 significant SNPs covering 20 genomic regions spread across the whole genome were identified (across stay-green scores and evaluations—E1, E2, and across the season) with *p*-values ranging from 10^–3^ to 10^–6^. Stay-green MTA analysis revealed hundreds of overlapping SNPs for the traits and seasons similar to QTL mapping studies. Significant MTAs for %GL7–49 traits were plotted based on *p*-values, in concentric circles (Figure 5). All significant SNP data are provided in Appendix A. Out of 6491 significant SNPs, 3996 were duplicates, and 2495 were unique SNPs.

We also found SBI-01 and SBI-09 had significant MTAs for the GL traits, and the significant probability values ranged from 9.96 × 10^–4^ to 7.62 × 10^–6^ and were annotated to identify the putative candidate genes. None of the identified significant MTAs were previously reported to be involved in any stay-green or associated mechanism of drought tolerance. On SBI-01, five unique significant MTAs were identified. Functional annotation of the SNP S1_12781217 (*Sobic.001G157700*) encoding for the Patatin-like phospholipase domain-containing protein, S1_18315751 (*Sobic.001G201800*) annotated as the *Os10g0503300* protein (HXXXD-type acyl transferase protein), and the remaining three SNPs viz., S1_12156535, S1_12564526, and S1_16647171 were in the intergenic region.

For SBI-09, four significant SNPs (MTAs) were detected, S9_ 58504227 was present in the intronic region of gene *Sobic.009G249800*, S9_58684354 (*Sobic.009G252100*), annotated as the predicted protein. S9_58757394 (*Sobic.009G253700*) was annotated as similar to the putative uncharacterized protein, and S9_58295230 was present in the intergenic region. SBI-01 and SBI-09 MTAs did not overlap with Haussmann et al. [21] stay-green QTLs [3]; nor were the candidate genes’ annotations relevant to any stay-green mechanism. 

The whole-genome SNPs count was compared with the targeted stay-green QTL region on SBI-10L with 96, 103, and 2296 unique significant SNPs in the 45–60 Mb region of SBI-10L, regions other than SBI-10 45–60 Mb, and the remaining genome of SBI-01 to SBI-09 (Figure 6A and Appendix A), respectively.

The significant SNPs functionally annotated on the SBI-10L 45–60 Mb regions, supported in cross-verifying the identification of candidate genes, with linkage map-based QTL analysis (and supported by MTA analysis) on SBI-10L for the stay-green QTLs [3] and the root angle QTL [20]. Annotation of 33 unique genes identified in the target regions showed a functional role in senescence, as well as drought stress mechanism (Appendix A and Figure 4). MTAs for weekly %GL scores on SBI-10L detected 169 stay-green-associated SNPs (*p*-values ranged from 9.99 × 10^−3^ to 3.00 × 10^−4^) present in the vicinity of 33 candidate genomic regions, among which a leucine-rich repeat protein (*Sobic.010G167500*), WRKY (*Sobic.010G205900*), Argonaute1-AGO1 (*Sobic.010G276600*), GA3 (*Sobic.010G172700*), and the NAM (*Sobic.010G273700*) genes appeared to be the other crucial candidate genes for delaying senescence on SBI-10L. Sixty-four candidate genes were identified by the CIM-based QTL mapping, and 33 by MTAs, of which 12 genes were commonly identified by both analyses (Appendix A, Figure 6B). In our current study, we used fine genetic mapping using informative SNPs to saturate the relatively large QTL interval. This resulted in the identification of several overlapping QTL clusters, along with putative candidate genes, supported by functional annotation of SNPs from MTA analysis.

### 3.8. Fine-Mapping and MTAs Revealed Putative Candidate Genes for the Stay-Green Trait on SBI-10L

A total of 203 putative candidate genes were identified based on fine-mapping, supported by the MTAs analysis, which revealed an analytical and biological significance for the genotype-phenotype relation. In all, 85 unique candidate genes were shortlisted, based on their functional annotation and their role in the biosynthetic pathways, spanning the genomic region across QTL clusters and corroborated with significant MTA (Appendix A; Figure 6B). Prioritized candidates such as AP2, WD40, NBS-LRR, LEA 2, Ca^+2^/calmodulin-dependent protein kinase (CAMK), Senescence-Associated Protein (SAP), No Apical Meristem (NAM), AGO1, WRKY, MADS-box transcription factors, Pentatricho Peptide Repeat (PPR) protein, Squamosa Binding Protein (SBP), Cytokinin-responsive GATA factor 1 (CGA1), and Zn finger protein were repeatedly identified in both E1, E2, and across season QTL analyses. These candidates will further help to understand the mechanism of post-flowering drought tolerance.

### 3.9. Stay-Green Candidate Gene Expression Profiling using Quantitative Real-Time PCR (qRT-PCR)

The putative candidates identified in the current study were further validated for their function by studying their expression profiling using qRT-PCR. The details of forward and reverse primers for the selected eight candidates are presented in the Table 6.

The Ct values of the actin gene were used to normalize the transcript levels of the designed primers. The candidate genes viz., CAMK, NAM, ARP, SAP, and the uncharacterized protein were upregulated (2.2, 4.8, 3.3, 2.8, and 1.5-folds, respectively), while AP2 and SBE were downregulated (2.1 and 4.0, respectively) between two parents, i.e., RSG04008-6 and J2614-11. These differential levels of expressions between parents of the identified candidate genes revealed their possible role in the stay-green mechanism (Figure 7). The other candidate gene, NBS-LRR, was a disease resistance protein, with inconclusive results of a minimal 0.2-fold change difference in the parents.

## 4. Discussion

Genetic dissection of the stay-green trait and elucidating the underlying mechanisms, will facilitate marker-assisted selection and help in developing terminal drought-tolerant sorghum cultivars. In the present study, we fine-mapped a stay-green QTL under terminal drought stress conditions that had a large interval of 45–60 Mb (15 Mb) of a genomic region and 72 cM of corresponding genetic distance on SBI-10L. In the present study, we identified 33 stay-green QTLs in the target recombination interval. Further, we fine-mapped this region into 7 QTL clusters and identified eight prioritized putative candidate genes, based on their annotations and biological pathways. Our study mainly focused on fine-mapping and validation of previously identified stay-green QTLs on SBI-10L [3], derived from E36-1 from an introgression line cross between RSG04008-6 × J2614-11. These introgression line-parents helped to minimize linkage drag for the transfer of QTLs into elite varieties [38]. Plant responses to drought stress depend on the intensity and duration of stress. Hence, individual seasons (E1 and E2) phenotyping data and across season data were utilized for the fine-mapping study. This study offers a new understanding of the role of fine-mapped QTL regions/genes involved in the stay-green mechanism of sorghum.

### 4.1. Stay-Green Fine-Mapping Population Influenced by the Environment and Its Effect on Yield

The genetic studies of F_2:4_ population with large F_2:4_ individuals assayed for the selection of homozygous recombinants and increased marker density, when combined with efficient experimental design (alpha lattice field design) usually helps improve the efficiency of QTL detection. Transgressive segregants for %GL were observed and could be of interest for breeding programs. The strong heritability nature of stay-green phenotype and tolerance to high temperatures with increased yield is a desirable trait for breeders and farmers. We observed that the mean grain-dry weight (GDW/plot) highly varied for both E1 and E2 (398 g/plot and 671 g/plot, respectively) (Table 1). This might be due to varied temperature and growth cycle regimes coinciding with different sowing dates (Jan2013-E1 and Nov2013-E2) (Appendix A and Appendix A). The strength of the correlation between %GL and GY depended on environmental conditions and genetic background. Moderately stressed plants were able to withstand drought and produced improved yields, showing positive relation between improved grain yield, with increased stay-green phenotype [39] (Appendix A).

### 4.2. The Significance of High-Resolution Genetic Linkage Mapping and Initial qRT-PCR-based Validation of Candidate Genes Associated with Senescence

SNP data obtained by GBS was efficient in identifying QTLs among F_2_ genotypes, using a recombinant selective genotyping approach, which is based on phenotypic variations. With the help of an integrated GBS–SNP–SSR map, a higher marker density along with a high correlation (r^2^ = 0.94) between genetic map order and physical map order was observed (Appendix A). This information, along with a large segregating population, enabled us to fine-map the stay-green genomic regions on SBI-10L.

Utilizing the field data from two different environments and across season data, 33 overlapping or co-localized QTLs (including previously mapped or new unique QTLs) on SBI-10L were identified. A similar region was also found to be associated with stay-green QTLs, near the flanking SSR markers Xgap001 and Xtxp141 on SBI-10L, and was mapped using two different populations that had a common stay-green donor E36-1 in several previous studies [3]. This QTL interval was also found to overlap with the nodal root angle QTL [20], which indicated the possible involvement of root traits in the stay-green mechanism. A plant height QTL (qPH10) was also overlapped in the stay-green fine-mapped region from the same source E36-1 [40]. This stay-green QTL co-localization with plant height QTL could be due to source-sink dynamics. Nineteen out of 33 QTLs were clustered into 7 genomic regions encompassing 503 kb (from 54388058bp-54891180bp) and 848 kb (from 59620328bp-60468750bp) interval region of SBI-10L. Remaining 14 QTLs were physically close to the QTL clusters, which revealed the possible involvement of the SBI-10L region in the stay-green expression. Identified SNPs in these QTL regions were further annotated. An annotated SNP (S10_60684319) in one such QTL, Q10GL14e_14 (with 10% PVE), was found to be present within the coding region of No Apical Meristem (NAM; as *Sobic.010G273700*) and was also near the starch branching (SBE) SNP S10_60695074 (*Sobic.010G273800*) candidate genes. NAM candidate was one of the NAC transcription factors that showed a differential expression of up to 4-fold upregulation (Figure 7) with RSG04008, when compared to J2614. NAC gene was reported to be associated with drought tolerance and senescence [19], but also acts as a negative regulator and increases grain nitrogen concentration in wheat [40].

QTL Q10GL14e_14 was stable across both environments and physically close to the cQstg10.5 cluster (containing 3 QTLs, i.e., Q10GL21a_13, Q10GL21a_14, Q10GL21a_across) with SNP S10_59620328 encoding LEA2 protein (*Sobic.010G259200*) involved in late embryogenesis and drought stress mechanisms [41]. QTL cluster cQstg10.6 found SNP S10_60059042 encoding Ca^+2^/calmodulin-dependent protein kinase (CAMK) (*Sobic.010G264400*) identified in both mapping and MTA analysis, and revealed differential expression in parents (2-fold upregulation in RSG04008 in comparison with J2614). This was reported to have a functional role in drought stress tolerance and senescence mechanism [42]. Similarly, SNP S10_60468746 (*Sobic.010G270300*) functional annotation was similar to SAP, which was located in the QTL cluster cQstg10.7 that expressed differentially with a 2-fold difference. AP2/ERF transcription factor, which was identified in cQstg10.1 was also reported to be involved in drought tolerance mechanism [19] and exhibited a 2-fold differential expression in parents.

Stay-green responsible genes also express under well-watered conditions. Therefore, the constitutive nature of stay-green is in correlation with drought tolerance, which might lead to more water availability under the grain filling stage, during drought stress conditions [43]. Expression of CAMK, NAM, SAP, AP2, SBE, and ankyrin repeat protein exhibited more than two-fold higher expression in drought tolerant genotype RSG04008-6, in comparison with senescent genotype J2614-11, suggesting their role in stay-green mechanism and in overcoming drought stress in sorghum. Most of these candidate genes are well-documented for their role in delayed senescence, and they were found to overlap within the identified stay-green QTL regions, indicating their possible role in drought tolerance in this population. This integration of genetic mapping, recombinant mapping, and significant MTAs involving high-density marker loci in the segregating F_2_ population, helped us to further fine-map this important region implicated in drought tolerance. This approach led to the identification of several SNPs present in the region, with known candidates and transcription factors (TFs) annotated and expressed during drought stress.

Over the last three decades, two major sources of stay-green sources, viz., B35 = IS 40606 and E36-1 = IS 30469, were used globally. Till date, there are no reports of fine-mapping of stay-green trait/QTL in E36-1, but attempts were also made by studying another popular stay-green source B35 on chromosomes other than SBI-10 [18].

### 4.3. Prioritization of Candidate Genes on SBI-10L

The functional annotation of candidates, co-localized with QTL clusters from fine-mapping studies and supported by MTA analysis, resulted in spotting putative candidate genes, SAP (S10_60468746; *Sobic.010G270300*), CAMK (S10_60059042; *Sobic.010G264400*), NBS–LRR protein (*Sobic.010G205600*), and WD40. The expression analysis of NBS–LRR revealed no variation between parents, indicating an inconclusive relationship with the senescence mechanism. In the case of SBI-01 (5MTA’s) and SBI-09 (4 MTA’s), significant SNPs were identified, but most of them, were intergenic or similar to predicted proteins and were not relevant to the senescence or the drought-tolerance mechanism. This could be attributed to 42.9% of unannotated sorghum transcriptome [28]. In total, 12 out of 85 significant MTAs overlapped with the candidate genes identified in the fine-mapped QTL clusters. *Sobic.001G157700* encoding for Patatin-like phospholipase domain containing protein is involved in the lipid metabolic process (GO: 0006629) and is also similar to the *Sobic.001G201800* gene- acyl tansferase activity, which had no role in the senescence mechanism.

The SNP S10_545269620 present in the gene *Sobic.010G202700* encodes an AP2-like transcription factor. Plants adapt to water deficiency by initiating a series of changes at the molecular, biochemical, and physiological pathways. Overexpression of AP2/ERF family genes are known to ameliorate stress tolerance under water-deficit conditions [19]. Four SNPs (S10_55270383, S10_55002960, S10_55618485, and S10_57692900) were annotated to the WRKY transcription factors (*Sobic.010G209200*, *Sobic.010G213200*, and *Sobic.010G234400*) and are reported to be involved in drought tolerance of rice [44]. Another SNP falling in the S10_59837691 MADS-box transcription factor (*Sobic.010G261800*) is known to be influenced under drought stress conditions, and alterations might lead to stay-green, which is involved in many developmental processes [45]. SNP S10_60557352 encoding exocyst subunit (*Sobic.010G271600*) was involved in autophagosome/autolysis, and thus it might be related to senescence [46]. Another SNP S10_60973291 encoding AGO1-related protein (*Sobic.010G276600*) is a component of the RISC complex, which degrades dsRNA and might be involved in the senescence mechanism. AGO1 transcriptional activity increases under ABA and drought treatments, consistent with the transcriptional elevation of MIR168a, is required for stabilizing AGO1 during stress response [47]. SNP S10_54891180 ankyrin repeat proteins (to which *Sobic.010G205800* is related) are reported to be involved during flower senescence, but in-depth investigations might reveal its role in leaf senescence [48]. The SNP S10_54593246 falls under the *Sobic.010G205900* gene that encodes the WD40/Transducin family proteins, reported to be involved in the senescence mechanism [19]. Penta tricopeptide repeat protein (PPR) (*Sobic.010G215400*-10_55836452, *Sobic.010G258600*- S10_59593663) is involved in the chlorophyll degradation mechanism by stabilizing the chloroplast RNA and participating in chlorophyll catabolism during senescence. However, alteration in PPR might lead to a stay-green phenotype [49]. SNP S10_55904321 encoding squamosa binding protein (SBP) family transcriptional factors (*Sobic.010G215700*) are involved in leaf, flower, fruit, and vegetative phase developmental stages [50]. Zinc finger proteins (*Sobic.010G243100*- SNP S10_58417069) are reported to be associated in senescence, and few MYB transcription factors in the senescence phenomenon [19]. The SNP S10_50865236 encoding CGA1 (CYTOKININ-RESPONSIVE GATA FACTOR 1; *Sobic.010G173300*) protein is implicated in the chloroplast metabolism. Phytohormones such as GA3 (*Sobic.010G172700*–SNP S10_50725358) and ethylene (C2H2), trigger the transcription factors associated with the senescence mechanism [19].

Many genes could be responsible for the stay-green trait, and it is interesting to note that the large effect stay-green QTL previously mapped onto the SBI-10L, appears to be highly complex. Not all of its components from donor parent E36-1 were necessarily economically or agronomically desirable. Therefore, detailed dissection and reconstruction studies were required to form a superior cassette of favorable alleles across this chromosome arm, which could be easily manipulated in an applied sorghum breeding program, targeting enhanced stability of high grain yield in drought-prone environments. Altogether 20 candidate genes were identified and their roles discussed in the stay-green mechanism, butthere is still a need to specify their role in different biochemical pathways and associated mechanisms. To address these questions, knock out mutations must be created, and Bulk Segregant Analysis (BSA) was carried out with a larger population that might help identify genes responsible for senescence or as distinct from stay-green. Expression analysis by qRT-PCR identified differentially expressed genes between the two parents, and further cloning of these genes helped unravel their functional role. The identified set of introgression lines in this population will help undertake a few of the above studies and develop a better understanding of the stay-green mechanism in sorghum.

## 5. Conclusions

The present study resulted in narrowing down the stay-green QTL (4 Mb) into 503 kb and 848 kb regions on SBI-10L, along with putative candidate genes involved in the drought tolerance mechanism. The increased temperature displayed negative effects on grain yield, when compared to medium temperature conditions, which exhibited significant increase in grain yields. Based on the SNP genotyping and expression profiling, putative candidate genes involved in the delayed senescence mechanism were identified from the fine-mapped region. This study could serve as a basis for generating transcriptomic data on stay-green, and gene-based mapping could be further advancement for confirming the role of candidate genes in stay-green expression. This study was the first to fine-map stay-green QTL with high resolution, providing the starting point for further in-depth studies and to identify candidate genes contributing to this important trait. This study emphasizes the challenge of linking QTL that modulates traits, to genes and pathways that affect trait expression. Further, genome editing or whole-genome sequencing of the parent along with Cg variants, could be identified for understanding their role in the stay-green phenotype and associated drought stress.

## Figures and Tables

**Figure 1 genes-11-01026-f001:**
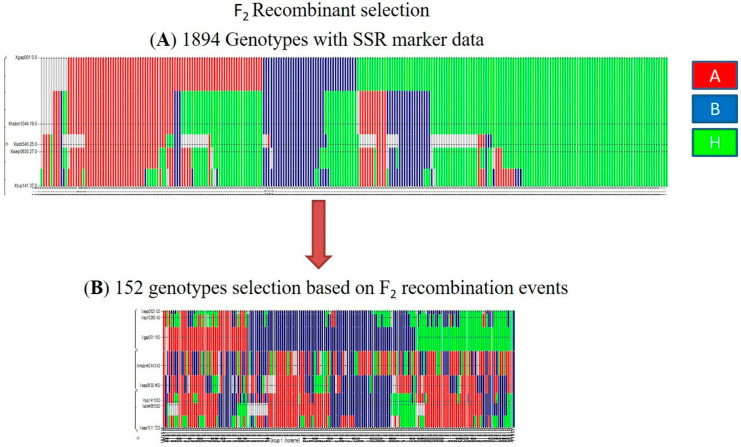
(**A**) F2 1894 fine-mapping population screened with 5 SSR markers within the QTL flanking region of SBI-10L, showing homozygous, heterozygous, and recombinant alleles. Allele ‘A’, homozygous for parent RSG04008-6, ‘B’ homozygous allele for parent J2614-11, and ‘H’ is a heterozygous allele from both parents. (**B**) Selected 152 recombinants based on the SSR markers data.

**Figure 2 genes-11-01026-f002:**
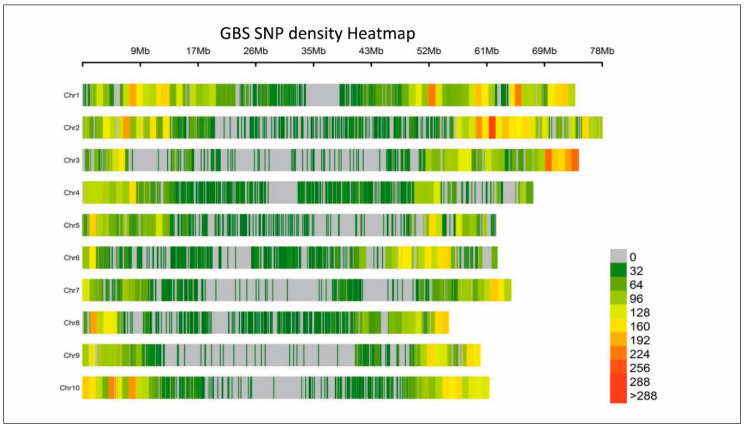
GBS–SNP density for cross RSG04008-6/J2614-11 developed for the stay-green QTL fine-mapping on SBI-10L. Genome-wide distribution of 29,506 SNPs represented in the color coded bars for the cross RSG04008-6/J2614-11 represented in 1 Mb window size. Green color sections across chromosome indicate low SNP density to red, being >288 SNPs per 1 Mb window size. Average SNP density (>150 SNP/Mb indicated by yellow color) was present in the telomeric regions.

**Figure 3 genes-11-01026-f003:**
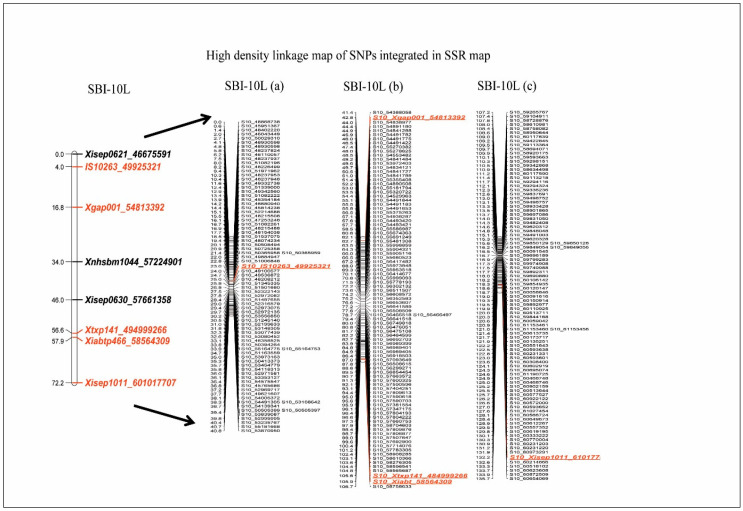
SBI-10L SSR map integrated with the horseshoe aligned SNP markers with linkage map distances estimated using the JoinMap.

**Figure 4 genes-11-01026-f004:**
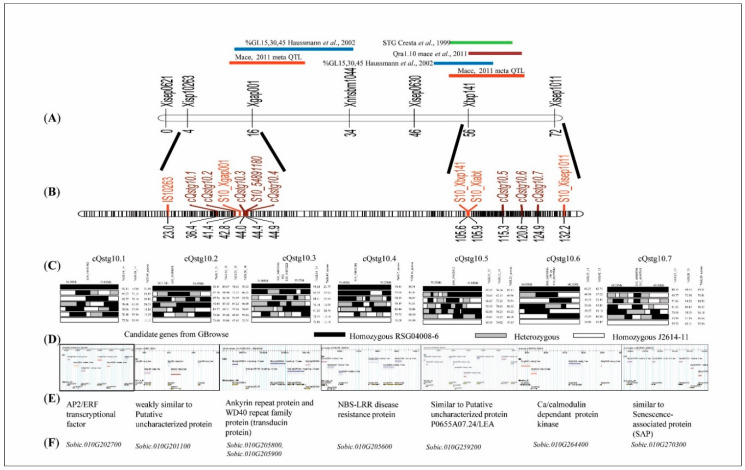
Fine-mapping of stay-green and GBrowse representation of candidate genes. (**A**) SSR map with overlapped stay-green and related QTLs from previous studies. (**B**) SSR integrated GBS–SNPs linkage map with clustered QTLs. (**C**) Across seasons percent green leaf area (%GL) data used to fine-map clustered stay-green QTLs and the putative candidate genes identified based on the variant SNPs. (**D**) Snapshot displaying G browse images showing the candidate genes present in the fine-mapped regions, (**E**) Functional annotation of the identified candidate genes. (**F**) Candidate gene ids for the *S**orghum bicolor* v3.1.

**Figure 5 genes-11-01026-f005:**
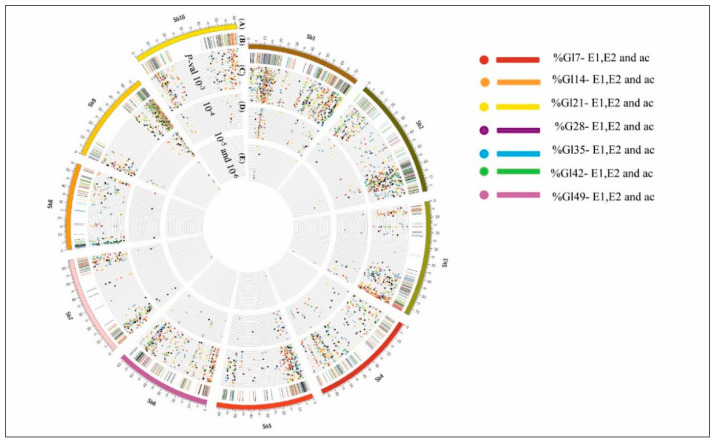
Stay-green trait significant MTAs in a circular plot from outside to inside aligned to the physical positions representing the (**A**) sorghum chromosomes (sb1-sb10). (**B**) Highlights for the significant SNPs overlapping regions for %GL, (**C**) significant MTAs scatter plot for the whole genome with *p*-value = 10^−3^. (**D**) Significant SNPs with *p*-value =10^−4^, and (**E**) significant MTAs SNPs with *p*-value = 10^−5^ & 10^−6^. Different %GL scores with red color (%GL7), orange color (%GL14), yellow color (%GL21), purple color (%GL28), blue color (%GL35), green color (%GL42), and pink color (%GL49).

**Figure 6 genes-11-01026-f006:**
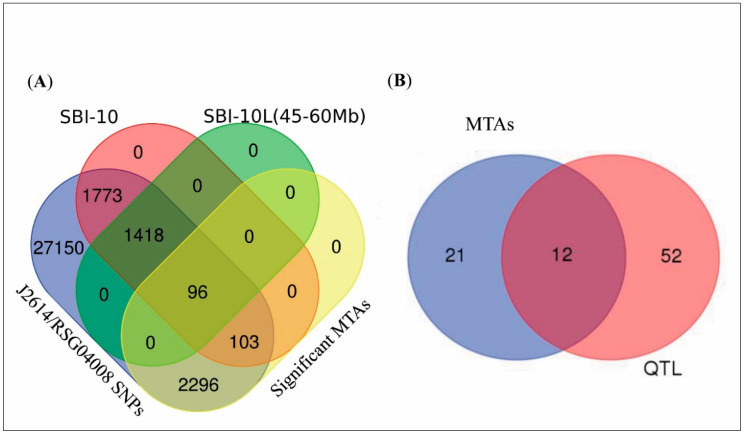
SNPs distribution on SBI-10L for the stay-green trait, chromosome, and genomic region. Venn diagram illustrating significant SNPs and common genes in MTAs and QTL studies for J2614/RSG04008-6. (**A**) Whole-genome scan SNPs, SBI-10 SNPs, SBI-10L SNPs, and the significant MTAs represented in the Venn diagram, and the (**B**) common genes identified by the stay-green MTAs and the QTL fine-map represented in the Venn diagram.

**Figure 7 genes-11-01026-f007:**
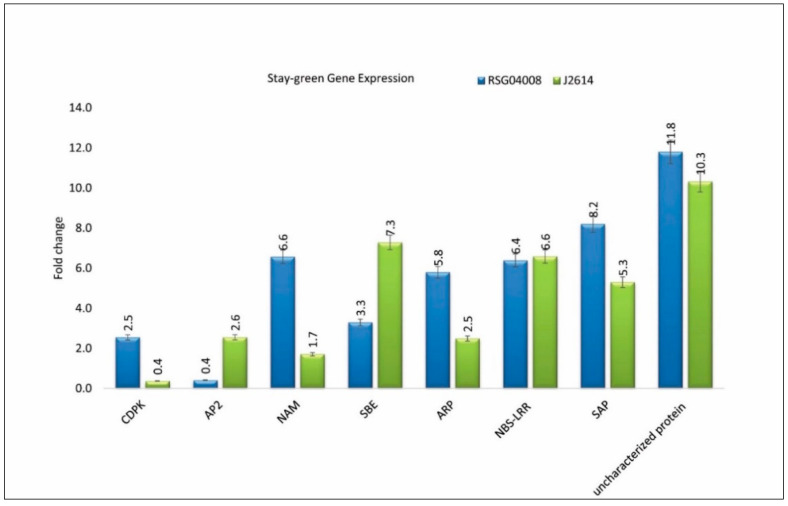
Relative changes of candidate gene expression in parents showing variant expression levels. CAMK-calcium/calmodulin-dependent protein kinase. AP2—Apetala 2, NAM—no apical meristem, ARP—ankyrin repeat protein, SBE—starch branching enzyme, NBS–LRR—disease resistance protein, SAP—senescence-associated protein, uncharacterized protein—putative uncharacterized protein.

**Table 1 genes-11-01026-t001:** Stay-green and grain-yield mean values.

S. No		Post Rainy 2012–2013/E1				Post Rainy 2013–2014/E2				Across Season			
		Mean			Range F_4_ Progeny	Mean			Range F_4_ Progeny	Mean			Range F_4_ Progeny
	Trait	RSG04008-6	F_4_ Progeny	J2614-11		RSG04008-6	F_4_ Progeny	J2614-11		RSG04008-6	F_4_ Progeny	J2614-11	
1	%GL 7 DAF	88.38	88.67	83.68	72.33–97.78	99.03	96.05	95.24	79.14–99.26	93.86	92.36	89.56	76.31–98.65
2	%GL14 DAF	79.18	79.58	72.06	58.73–96.64	84.22	84.29	85.26	68.60–96.93	81.71	81.94	78.65	63.34–96.69
3	%GL21 DAF	62.52	65.96	61.88	43.29–93.77	74.39	73.73	75.05	63.34–84.91	68.43	69.85	68.48	54.12–90.37
4	%GL28 DAF	47.00	50.72	47.68	29.92–78.11	66.49	65.03	64.04	51.36–77.49	56.83	57.88	55.78	42.44–77.36
5	%GL35 DAF	34.46	39.49	37.72	22.90–62.25	54.17	51.57	50.54	19.89–66.05	44.39	45.53	44.19	19.74–61.07
6	%GL42 DAF	23.44	29.41	30.73	17.19–45.88	25.96	39.73	40.1	6.09–53.89	25.28	34.71	35.25	13.73–48.77
7	%GL 49 DAF	17.12	18.56	20.33	9.90–31.90	11.68	29.07	28.4	4.73–44.61	14.67	24.01	24.94	8.72–38.12
8	PnDW/plot	598.1	577.49	372.53	416.17–779.52	945.95	935.88	1016.03	763.31–1102.78	768.65	756.15	738.23	668.70–887.70
9	GDW/plot	392.72	397.96	233.42	267.61–539.88	632.53	650.60	732.21	531.50–777.47	513.47	523.70	518.19	461.19–599.05
10	PHI	66.9	68.4	62.88	63.08–78.21	68.63	70.14	71.69	66.35–74.21	67.48	69.27	67.72	65.00–73.43
11	HGM	2.14	1.94	2.06	1.28–2.44	3.28	2.95	2.71	2.27–3.73	2.70	2.45	2.38	1.79–3.07
12	GNP/plot	20,456.12	21,232.45	17,807.87	18,453.82–43,294.30	20,212.25	22,191.42	26,891.54	17,204.13–27,250.23	20,746.04	21,720.92	21,254.12	19,611.06–33,652.47
13	GNPP	786.78	816.64	684.92	709.77–1665.17	777.40	853.52	1034.29	661.70–1048.09	797.92	835.42	817.47	754.27–1294.33

%GL—percentage green leaf area (7–49) days after flowering for E1 (13) & E2 (14), PnDW/plot—panicle dry weight per plot, GDW/plot—grain dry weight per plot, PHI—panicle harvest index, GNP/plot—grain number per plot, GNPP—grain number per panicle. BLUPs for all traits were significant with *p* value ≤ 0.05, except for %GL42 and 49.

**Table 2 genes-11-01026-t002:** Genotype variance, genotype × environment, standard error, and heritability estimates (on the mean basis) for stay-green scores derived from cross RSG04008-6 × J2614-11.

		Post Rainy 2012–2013			Post Rainy 2013–2014			Across Season			
S. No.	Trait	σ^2^g	±SE	h^2^	σ^2^g	±SE	h^2^	σ^2^g	GхE	±SE	h^2^
1	%GL7 DAF	64.34	7.941	75.38	26.42	5.92	69.37	73.44	100.99	7.00	81.80
2	%GL14 DAF	183.30	13.8	74.27	42.63	7.65	68.64	181.90	256.50	11.16	81.42
3	%GL21 DAF	198.33	13.85	75.63	17.13	6.54	54.59	160.13	283.30	10.83	80.39
4	%GL28 DAF	121.70	11.13	74.65	30.21	7.31	62.93	120.54	182.82	9.42	80.31
5	%GL35 DAF	63.26	8.775	71.14	62.33	9.48	67.52	98.61	164.42	9.14	77.99
6	%GL42 DAF	29.74	8.04	57.98	97.99	8.39	80.70	80.03	210.67	8.21	78.08
7	%GL49 DAF	28.20	8.719	52.67	104.33	10.13	75.31	79.26	249.26	9.44	72.73
8	PnDW/plot	4307.00	4562.00	57.94	8033.33	208.50	35.66	7717.67	40,344.00	162.40	46.74
9	GDW/plot	3341.67	88.51	56.13	3906.67	139.50	37.59	3610.33	21672.00	113.40	45.70
10	PHI	9.48	7.61	32.96	1.73	8.02	7.45	8.75	69.66	7.65	30.99
11	HGM	0.04	0.12	87.95	0.07	0.26	75.91	0.08	0.11	0.19	86.91
12	GNP/plot	12,233,333.00	1,282,200	18.25	6,710,000.00	5443.00	40.45	12,460,000.00	116,400,000.00 ^ns^	9850	27.81
13	GNPP	18,072.00	493.20	18.23	9924.00	209.40	40.45	18417.00	172259.00 ^ns^	378.80	27.78

All genetic effects significant at *p* = 0.001 significance, except for the ns σ^2^g—genotypic variance, ±SE—standard error, h^2^—heritability, G × E—genotype environment interaction, %GL—percentage green leaf area, and DAF—days after flowering.

**Table 3 genes-11-01026-t003:** Correlation between stay-green weekly scores with grain yield for E1 & E2.

Trait	%GL_7	%GL_14	%GL_21	%GL_28	%GL_35	%GL_42	%GL_49	GDW	GDW/Plot	GNPP	GNP/Plot	HGM	PHI	PnDW	PnDW/Plot
%GL13_7	1														
%GL14_7	1														
%GL13_14	0.98 **	1													
%GL14_14	0.68 **	1													
%GL13_21	0.93 **	0.94**	1												
%GL14_21	0.65 **	0.88 **	1												
%GL13_28	0.89 **	0.9 **	0.99 **	1											
%GL14_28	0.72 **	0.82 **	0.88 **	1											
%GL13_35	0.82 **	0.82 **	0.93 **	0.97 **	1										
%GL14_35	0.67 **	0.81 **	0.76 **	0.75 **	1										
%GL13_42	0.68 **	0.66 **	0.81 **	0.87 **	0.94 **	1									
%GL14_42	0.5 **	0.59 **	0.56 **	0.53 **	0.77 **	1									
%GL13_49	0.81 **	0.83 **	0.85 **	0.87 **	0.87 **	0.85 **	1								
%GL14_49	0.65 **	0.61 **	0.54 **	0.6 **	0.7 **	0.82 **	1								
GDW_13	0.08	0.1	−0.02	−0.06	−0.13	−0.22	−0.08	1							
GDW_14	0.12	0.01	−0.1	−0.09	0.06	0.12	0.15	1							
GDW_plot_13	0.01	−0.02	−0.08	−0.08	−0.09	−0.11	−0.07	0.75 **	1						
GDW_plot_14	0.09	0.06	0.01	0.01	0.06	0.09	0.1	0.57 **	1						
GNPP_13	0.1	0.07	0.06	0.05	0.05	0.05	0.01	0.01	0.05	1					
GNPP_14	0.02	0.06	0.01	0.03	0.09	−0.02	−0.04	0.04	0.02	1					
GNP_plot_13	0.07	0.07	0.06	0.05	0.05	0.05	0.01	0.01	0.05	1	1				
GNP_plot_14	0.09	0.06	0.01	0.01	0.06	0.09	0.1	0.57 **	1	0.02	1				
HGM_13	−0.16	−0.18	−0.17	−0.11	−0.07	0.05	−0.01	−0.04	0.15	−0.13	−0.13	1			
HGM_14	−0.09	−0.03	−0.09	−0.13	0.01	−0.01	−0.08	−0.12	0.06	0.01	0.06	1			
PHI_13	−0.05	−0.07	−0.11	−0.12	−0.12	−0.13	−0.07	0.62 **	0.62 **	−0.02	−0.02	−0.15	1		
PHI_14	−0.02	0.05	0.03	0.03	0.04	0.05	0.07	0.33 **	0.22 **	−0.12	0.22 **	−0.22 **	1		
PnDW_13	0.11	0.15	0.03	−0.02	−0.11	−0.22	−0.07	0.95 **	0.65 **	0.02	0.02	0.01	0.36 **	1	
PnDW_14	0.12	−0.02	−0.12	−0.12	0.05	0.11	0.12	0.91 **	0.51 **	0.09	0.51 **	−0.05	−0.1	1	
PnDW_plot_13	0.04	0.01	−0.05	−0.05	−0.07	−0.08	−0.06	0.65 **	0.95 **	0.08	0.08	0.22 **	0.35 **	0.65 **	1
PnDW_plot_14	0.08	0.03	−0.01	−0.01	0.04	0.07	0.05	0.38 **	0.88 **	0.07	0.88 **	0.17	−0.28	0.53 **	1

** ≥ highly significance PnDW/plot_13/14—panicle dry weight per plot for E1/E2, GDW/plot_13/14—grain dry weight per plot E1/E2, PHI_13/14—panicle harvest index E1/E2, GNP/plot_13/14—grain number per plot E1/E2, GNPP_13/14—grain number per panicle E1/E2, %GL—percentage green leaf area, 7–49 days after flowering for E1 (13) and E2 (14). E1 is a normal font, and E2 is in italics font.

**Table 4 genes-11-01026-t004:** Stay-green QTL mapping on SBI-10L for post-rainy 2013 (13), 2014 (14), and across the season.

QTLs on SBI-10	Pos cM	Nearest Marker	Marker Interval	Support Interval	LOD	%R^2^ (PVE)	Additive	Dominant
Q10GL7a_across	44.41	S10_54891180	Xgap001_54813392–S10_54491782	42.8–45.9/44.9	2.76	6.75	2.2331	0.7273
Q10GL7a_13	41.41	S10_54388058	S10_55181668–Xgap001_54813392	40.6–42.4	2.93	8.86	2.8201	−2.6178
Q10GL7a_14	104.81	S10_58565687	S10_58276305–Xiabt_58564309	103.7–105.9	3.40	6.98	−1.5675	−3.095
Q10GL7b_14	112.11	S10_59294324	S10_58634498–S10_59498752	110.9–113.1	3.11	9.43	−1.5537	1.6478
Q10GL7combined r^2^						32.02		
Q10GL14d_14	29.01	S10_52316579	S10_51497655‒S10_52973075	28.4‒29.4	2.21	0.09	0.5767	5.886
Q10GL14a_14	36.41	S10_54575547	S10_54575547–S10_52969717	36.3–37.2	2.59	5.03	2.0097	2.8007
Q10GL14b_14	45.01	S10_54841288	Xgap001_54813392–S10_55270382	42.8–47.4	2.55	6.49	2.7621	0.5624
Q10GL14e_14	123.61	S10_60308400	S10_60231331‒S10_60602919	122.9‒124.1	2.37	9.70	−2.3214	3.225
Q10GL14c_14	129.51	S10_60557352	S10_60612267–S10_60333222	129–130.0	3.38	8.94	−2.7106	6.1702
Q10GL14a_13	41.41	S10_54388058	S10_55181668–Xgap001_54813392	40.7–42.8	2.86	9.00	4.9036	−4.2892
Q10GL14a_across	44.41	S10_54891180	Xgap001_54813392–S10_54491782	42.8–45.6	3.73	10.20	4.5978	0.0381
Q10GL14b_13	82.71	S10_56692703	S10_56476051‒S10_56989405	82.6‒85	2.18	4.24	−5.2408	−4.2484
Q10GL14combined r^2^						53.68		
Q10GL21a_across	115.31	S10_59620328	S10_59620312–S10_59850129	114.5–115.6	2.91	10.14	−5.0744	2.4482
Q10GL21b_13	41.41	S10_54388058	S10_55181668‒Xgap001_54813392	40.5‒42.8	2.18	6.91	4.9353	−3.5171
Q10GL21a_13	115.31	S10_59620328	S10_59620312–S10_59849054	114.5–116.2	2.76	9.07	−6.0311	6.0466
Q10GL21c_13	125.01	S10_60468748	S10_60308400–S10_60513644	123.6–125.5	2.33	1.41	−0.3211	13.282
Q10GL21b_14	79.41	S10_56641518	S10_56608572‒S10_56692703	74.3‒82.7	2.12	7.04	−1.4956	2.3253
Q10GL21c_14	99.01	S10_57507647	S10_58704603‒S10_57714076	97.9‒100.4	2.16	5.31	1.055	−4.1377
Q10GL21a_14	115.31	S10_59620328	S10_59620312–S10_59850129	114.5–115.6	2.74	9.20	−2.2148	1.2957
Q10GL21combined r^2^						49.08		
Q10GL28a_14	36.41	S10_54575547	S10_54575547–S10_52969717	36.3–37.2	3.67	4.96	2.044	3.4449
Q10GL28b_14	41.41	S10_54388058	S10_52995995–Xgap001_54813392	39.8–42.8	2.77	6.62	2.1001	0.7883
Q10GL28a_across	124.91	S10_60468748	S10_60308400–S10_60513644	123.6–125.5	2.74	2.85	−0.6203	7.2213
Q10GL28a_13	125.01	S10_60468748	S10_60308400–S10_60513644	123.6–125.5	2.52	1.20	−0.1975	11.128
Q10GL28combined r^2^						15.62		
Q10GL35a_13	121.91	S10_60135251	S10_60613735‒S10_60593638	121.4‒122.5	2.36	2.06	−0.6135	8.0863
Q10GL42a_13	121.91	S10_60135251	S10_60173717–S10_60593638	121.7–122.5	2.59	4.54	−0.5686	5.187
Q10GL42b_13	131.91	S10_60973291	S10_60231220‒Xisep1011_61017707	131.2–132	2.01	0.52	2.4162	3.149
Q10GL42d_14	32.31	S10_53077439	S10_52199633‒S10_55164775	30.8‒33.9	2.47	6.66	2.0509	−6.9547
Q10GL42b_14	38.41	S10_53108642	S10_52969717‒S10_53839087	37.7‒39.6	2.33	2.28	3.5114	4.965
Q10GL42a_14	102.31	S10_57783305	S10_57714076–S10_58610366	100.4–103.6	2.60	2.43	1.7652	−29.678
Q10GL42c_14	107.81	S10_58728876	S10_58758633‒S10_58758082	106.7‒108.5	2.44	5.75	2.0457	−7.1224
Q10GL42combined r^2^						22.18		
Q10GL49a_13	34.71	S10_51163559	S10_50394264‒S10_54118313	33.9–35.8	2.37	2.17	1.4224	3.2659
Q10GL49a_across	36.41	S10_54575547	S10_54575547–S10_52969712	36.4–37.2	2.54	2.45	2.1417	3.2671
Q10GL49b_across	45.01	S10_54841288	Xgap001_54813392–S10_54491782	42.8–45.9	3.04	4.08	2.6273	2.1788
Q10GL49combined r^2^						8.71		

QTL—quantitative trait loci, Pos—position of QTL in cM, LOD—logarithm of odds, r^2^%—percentage of phenotypic variance Add—Additive, and Dom—Dominance.

**Table 5 genes-11-01026-t005:** Details for the stay-green QTL cluster analysis.

cQTL	Nearest Marker	Position cM	Marker Intervals	No.of QTLs	Individual QTLs Co-Localized	Gene ID/MTAs	Combined *r^2^*	Previous Studies Reporting stg QTL	Candidate Genes	SNP Effect
*cQstg10.1*	S10_54553482	36.4	36.4	3	*Q10GL14a_14, Q10GL28a_14, Q10GL49a_across*	*Sobic.010G202700*	12.4	Haussmann et al. 2002 [3]	*AP2/ERF transcriptional factor*	Splice_site_ region + Intron
*cQstg10.2*	S10_54388058	41.4	39.8–42.8	4	*Q10GL7a_13, Q10GL14a_13, Q10GL21b_13, Q10GL28b_14*	*Sobic.010G201100*	31.3	Haussmann et al. 2002 [3]	*weakly similar to Putative uncharacterized protein*	Non synonymous
*cQstg10.3*	S10_54891180S10_54899228	44.4	54.6–54.6	2	*Q10GL14b_14, Q10GL49b_across*	*Sobic.010G205800, Sobic.010G205900*	10.6	Haussmann et al. 2002 [3]	*Ankyrin repeat protein and WD40 repeat family protein (transducin protein)*	Intron and synonymous
*cQstg10.4*	S10_54841288	45.0	42.8–47.4	2	*Q10GL7a_across, Q10GL14a_across*	*Sobic.010G205600*	16.9	Haussmann et al. 2002 [3]	*NBS-LRR disease resistance protein*	synonymous
*cQstg10.5*	S10_59620328	115.3	114.5–115.6	3	*Q10GL21a_13, Q10GL21a_14, Q10GL21a_across*	*Sobic.010G259200*	28	Haussmann et al. 2002 [3]	*similar to Putative uncharacterized protein P0655A07.24/LEA*	Nonsynonymous
*cQstg10.6*	S10_60059042	121.9	121.4‒122.5	2	*Q10GL35a_13, Q10GL42a_13*	*Sobic.010G264400*	6.6	Haussmann et al. 2002 [3]	*Ca/calmodulin dependent protein kinase*	Intron
*cQstg10.7*	S10_60468746	125.0	124.7–125.5	3	*Q10GL21c_13, Q10GL28a_13, Q10GL28a_across*	*Sobic.010G270300*	5.5	Haussmann et al. 2002 [3]	*similar to Senescence-associated protein*	synonymous

cQTL—cluster quantitative trait loci, cM—CentiMorgan, MTAs—marker trait associations, SNP—single nucleotide polymorphism.

**Table 6 genes-11-01026-t006:** Primers designed for candidate gene expression profiling.

	SNP ID	Gene ID 3.1V	Gene Abbrevation	Gene Function	Forward Primer Seq	Reverse Primer Seq	Product	Allele
1	S10_54553482	*Sobic.010G202700*	AP2	similar to Apetala 2	attcgacactgctcatgctg	gtacttggagctgcctctgg	196	T/A
2	S10_54841288	*Sobic.010G205600*	NBS-LRR	*NBS-LRR disease resistance protein*	ggagtgcagcattgttcaga	caatgagctcaggggcttag	184	A/G
3	S10_54891180	*Sobic.010G205800*	ARP	similar to ankyrin repeat-containing protein-like	cgagcatggagcagacataa	tgtgtcgcctgatacccata	153	A/T
4	S10_59620328	*Sobic.010G259200*	Uncharacterised	similar to Putative uncharacterized protein P0655A07.24	acctgctgtacaagcccaag	tcggtcttggagttggagtt	142	A/T
5	S10_60059042	*Sobic.010G264400*	CAMK	*Calcium/calmodulin-dependent protein kinase*	gtgtggcagtaggccatttt	tgcttggacagcagtcattc	175	A/C
6	S10_60468746	*Sobic.010G270300*	SAP	similar to Senescence-associated protein	ggctcaggaatgacgaaaaa	cagctcattctccctccaag	192	C/A
7	S10_60684319	*Sobic.010G273700*	NAM	*No apical meristem (NAM) protein*	ggagtggagaccatgacgat	gtcagaaatgggtcctgcat	182	T/A
8	S10_60695074	*Sobic.010G273800*	SBE	*Starch branching enzyme I precursor*	attgggatcctcctgcttct	ccatcaactgaacggtgttg	192	A/G

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
