# Peer review of "Fine-Mapping of Sorghum Stay-Green QTL on Chromosome10 Revealed Genes Associated with Delayed Senescence"

_genes, 2020, doi:10.3390/genes11091026_

Round 1
Reviewer 1 Report
- Section 2.12
the use of GAPIT for the determination of marker trait associations has been used incorrectly and, results generated from this are not justifiable even though we have 12 candidate genes overlapping between the two mapping methods.
The population used in this study is a family-based population and not suitable for GAPIT.
The basic scenario in GWAS is to calculate the association between each marker and a phenotype of interest that has been scored across unrelated lines/individuals (unrelated individuals means distantly related and heterogeneous individuals) of a diverse collection.
The EMMA algorithm requires a population structure and kinship. How was this estimated from the family-based populations used in this study?.
2. Section 2.13
In section 2.13, total RNA was extracted from 7- day old seedlings. The seedlings were not subjected to any drought stress. The results show that the tested genes are constitutively expressed. The idea that they are upregulated or downregulated cannot be determined based on this study (ln: 544-550). A more structured experiment under stress is needed.
Ln: 550-552. This statement cannot be justified from this study “Two candidate genes not differentiated clearly in the two parents shows their reduced role in stay green mechanism compared with other candidate genes (Figure 7). Again, a more structured experiment under some form of stress is required to justify these results.
Author Response
- Section 2.12
The use of GAPIT for the determination of marker trait associations has been used incorrectly and, results generated from this are not justifiable even though we have 12 candidate genes overlapping between the two mapping methods.
The population used in this study is a family-based population and not suitable for GAPIT.
The basic scenario in GWAS is to calculate the association between each marker and a phenotype of interest that has been scored across unrelated lines/individuals (unrelated individuals means distantly related and heterogeneous individuals) of a diverse collection.
The EMMA algorithm requires a population structure and kinship. How was this estimated from the family-based populations used in this study?
Authors Response:
The authors agree to this observation. Primary focus of this current study is to genetically fine-map stay-green QTL region on SBI-10L, which is still largely unexplored. Towards this we have primarily used informative SNPs, filtered from GBS data, to saturate the relatively large QTL interval. This QTL mapping study resulted in the identification of several overlapping QTL clusters. To support these findings and to confirm the informative SNP identification, we conducted the GWAS on this F2:F4 population, with the MLM (Q+K) method in GAPIT to remove false positive in identifying significant marker-trait associations. GWAS provided support/corroboration of short-listed candidates identified in QTL clusters and was only a support study. Also, we wish to inform that there are several GWAS studies based on bi-parental populations such as Kale et al., 2015 (Scientific Reports doi: 10.1038/srep15296) and Wu et al., 2016 (Plant Biotechnology Journal doi: 10.1111/pbi.12519). Line numbers: 227-228, 232-233
This, we have suggested following changes to indicate the supporting nature of GWAS and MTAs in the draft and highlight the role of filtered GBS-SNPs falling in each QTL cluster for candidate gene identification –
- Abstract: Page 1, Line 34-35
- Section 12 Marker trait associations (MTAs) and line numbers : 228-229, 234-236
- Added a new section 13 Candidate genes identification. line number: 241
- Section 3.7. MTAs for stay-green traits : line 543 – 547
- Changed the section 4.3. ‘MTA analysis and prioritization of candidate genes on SBI-10L’ to ‘3. Prioritization of candidate genes on SBI-10L’; moved the section from line no. 694-704 to line no. 708-717 to bring continuity.
- Section 2.13
In section 2.13, total RNA was extracted from 7- day old seedlings. The seedlings were not subjected to any drought stress. The results show that the tested genes are constitutively expressed. The idea that they are upregulated or downregulated cannot be determined based on this study (ln: 544-550). A more structured experiment under stress is needed.
Author’s response:
The present study main aim is to fine map the stay-green QTL region and identify the candidate genes. We reached the target by saturating the target QTL region with GBS-based informative SNPs and fine mapped the region in to two regions- one with 503kb (54388058bp- 54891180bp) and the other with 848kb region (59620328bp- 60468750bp). To further support these results, we planned expression studies with limited number of contrasting samples with due diligence about quality (3 biological replicates and 2 technical replicates for each sample) was conducted as both the parents are showing very strong phenotypic variation (supplementary Figure1). Stay-green is constitutive in nature and able to express in between parents without drought stress which may lead to more water availability under grain filling under drought stress conditions [43] (Burke et al., 2010, doi: 10.2134/agronj2009.0465). Most of this study was a supporting experiment with very smaller funding from the Department of Science and Technology (DST) – Science and engineering research board (SERB) project funded by Govt. of India. Line numbers: 677-679
While authors agree to the suggestion of a conclusive study with proper stress experiment set-up, the scope and resources (including time, budgets) will form a basis for a separate project. The current COVID-19 spread and restrictions on normal/daily life movements is a further limitation. We believe that the limited gene expression studies and protein fold changes reported in the current study itself is a very positive and significant output from our study, which will add significantly to public domain knowledge about this QTL region unavailable till date.
Ln: 550-552. This statement cannot be justified from this study “Two candidate genes not differentiated clearly in the two parent’s shows their reduced role in stay green mechanism compared with other candidate genes (Figure 7). Again, a more structured experiment under some form of stress is required to justify these results.
Author’s response: Authors agree completely. We have now changed the section to report the finding about the specific candidates in more realistic way. Please refer to line no 574-575 under Results section -
“The other candidate gene, NBS-LRR, is a disease resistance protein, with inconclusive results of a minimal 0.2 fold change difference in the parents”
Also, refer to section 4.3. Prioritization of candidate genes on SBI-10L line no. 708-709
“The expression analysis of NBS-LRR revealed no variation between parents indicating inconclusive relationship with the senescence mechanism”
Reviewer 2 Report
The review of a manuscript titled “Fine mapping of sorghum stay-green QTL on chromosome 10 revealed genes associated with delayed senescence” by K. N. S. Usha Kiranmayee, C. Tom Hash, Sivasubramani Selvanayagam, Ramu Punna, A. Bhanu Prakash, Abhishek Rathore, P. B. Kavi Kishor, Rajeev Gupta, Santosh P. Deshpande
The manuscript by Santosh P. Deshpande (corresponding author) and others aimed at studying the sorghum stay-green trait QTL region on chromosome 10 contributing to post-flowering drought tolerance of this plant species. Based on the expression profiling and SNP genotyping, the Authors identified putative candidate genes involved in delayed leaf senescence and drought tolerance in fine mapped region. The manuscript is well-written, and the results of the experiment are interesting; however, it contains several deficiencies that require attention and should be corrected.
Abstract and Introduction
The Abstract contains the aim of experiments and most important conclusions. The Introduction section presents important and actual state of knowledge in the manuscript subject. It presents the basic features and economic importance of the species under study, the basic aspects of the leaf senescence process and the molecular relationship between leaf senescence and drought tolerance.
The Materials and Methods section presents necessary information connected with biological material, including the parents and progeny lines.
The Authors describe the F2 population screening with 8 SSRs (Line 94-97). Please characterize these five SSRs and three additional SSRs or cite the appropriate references,
Figure 1: I think the abbreviations A, B, and H should be explained in the Figure caption (Line: 104-106),
The Results
Line 297-305: the Authors refer to Figure S1, which presents %GL values for parents and progeny lines. I think it would be beneficial to introduce SE bars or/and letters indicating homogeneous groups in these Figures. I also think that Figure S9 (which presents the changes of %GL) should be placed after Figure S1 in Supplementary Figures. It will enable the rate of leaf senescence in parents and progeny to be assessed. The part of the Discussion section (Line: 572-577) referred to Figure S9 should be also moved to the Result section (after the line 305).
Discussion
The beginning part of the Discussion (Lines: 572-576, 581-583, 585-591) contains data that should be moved to the Results section. In my opinion, this part of the Discussion section should be rebuilt, and its relevant part moved to the Results.
Supplementary material
Figure S2: there is no X-axis title, there is no P1 and P2 description,
Figure S3: please unify %GLA and %GL abbreviation, please add a better title to each of figure
Editing errors:
Line 82: and putative change to and putative
Line 131: per cent change to percent
Line 250: 620C change to 62ºC
Table 1: please unify the abbreviations in the table and under the table, for example PHI and PHI_13/14, also GDW/plot or GWD/plot
Line 508: regionse change to regions
Line 652: to identify – delete, repetition
References:
Line 783: Sorghum change to Sorghum
Line 797: Oryza sativa change to Oryza sativa
Line 803: Sorghum bicolor change to Sorghum bicolor
Line 806: J plant change to J Plant
Line 821: Sorghum bicolor change to Sorghum bicolor
Line 827: Sorghum bicolor change to Sorghum bicolor
Line 835: Sorghum bicolor change to Sorghum bicolor
Line 839: Sorghum bicolor change to Sorghum bicolor
Line 854: Sorghum bicolor change to Sorghum bicolor
Line 860: Drosophila melanogaster change to Drosophila melanogaster
Line 883: Sorghum bicolor change to Sorghum bicolor
Author Response
The Authors describe the F2 population screening with 8 SSRs (Line 94-97). Please characterize these five SSRs and three additional SSRs or cite the appropriate references,
Authors response: Characterization and fine mapping population development was explained in detail in reference manuscript 21 (Kiranmayee et al., 2016). So cited appropriate reference [21]. Line number: 98
Figure 1: I think the abbreviations A, B, and H should be explained in the Figure caption (Line: 104-106)
Authors response :Abbreviations included for A, B, and H. Allele ‘A’ homozygous for parent RSG04008-6 , ‘B’ homozygous allele for parent J2614-11 and ‘H’ is heterozygous allele from both the parents. Line numbers:106-108
The Results
Line 297-305: the Authors refer to Figure S1, which presents %GL values for parents and progeny lines. I think it would be beneficial to introduce SE bars or/and letters indicating homogeneous groups in these Figures.
I also think that Figure S9 (which presents the changes of %GL) should be placed after Figure S1 in Supplementary Figures. It will enable the rate of leaf senescence in parents and progeny to be assessed. The part of the Discussion section (Line: 572-577) referred to Figure S9 should be also moved to the Result section (after the line 305).
Author’s response: According to reviewers suggestions we have modified the supplementary figure S1 and included standard error bars.
Figure S9 was also moved and now renamed to S2 and the mentioned lines was moved from discussion section to the results section. Line numbers: 309-314, 320-322, 332-337
Discussion
The beginning part of the Discussion (Lines: 572-576, 581-583, 585-591) contains data that should be moved to the Results section. In my opinion, this part of the Discussion section should be rebuilt, and its relevant part moved to the Results.
Author’s response: the mention sections were moved to results (Line numbers: 309-314, 320-322, 332-337) and the disturbed discussion section was rephrased. Line numbers: 608-617
Transgressive segregants for %GL were observed and could be of interest for breeding programs. The strong heritability nature of stay-green phenotype and tolerance to high temperatures with increased yield is desirable trait for breeders and farmers. We observed mean grain dry weight (GDW/plot) was highly varied for both E1 and E2 (398 g/plot and 671 g/plot, respectively) (Table 1). This might be due to varied temperature and coinciding growth cycle regimes coinciding with different sowing dates (Jan2013-E1 and Nov2013-E2) (Supplementary Table S4 and Supplementary Figure S3). The strength of the correlation between % GL and GY depends on environmental conditions and genetic background. Moderately stressed plants were able to withstand drought and produced improved yields showing positive relation between improved grain with increased stay-green phenotype [39] (Supplementary Figure S3).
Supplementary material
Figure S2: there is no X-axis title, there is no P1 and P2 description,
Author’s response: Agreed with reviewers. Edited Figure S2 given X-axis title %GL and described P1=RSG04008-6 and P2=J2614-11
Figure S3: please unify %GLA and %GL abbreviation, please add a better title to each of figure
Author’s response: Edited %GLA to %GL in supplementary figures
Edited titles of figures as -
“Supplementary Figure S4 – Phenotype distribution for Percent green leaf area (%GL) E1, E2 season data, P1=RSG04008-6, P2=J2614-11”
“Supplementary Figure S5 –phenotype distribution for Percent green leaf area (%GL) across season data, P1=RSG04008-6, P2=J2614-11”
Editing errors:
Line 82: and putative change to and putative
Author’s response: Edited. Line numbers: 83
Line 131: per cent change to percent
Author’s response: Edited. Line numbers: 139
Line 250: 620C change to 62ºC
Author’s response: Edited. Line numbers: 259
Table 1: please unify the abbreviations in the table and under the table, for example PHI and PHI_13/14, also GDW/plot or GWD/plot
Author’s response: Edited. PnDW/plot-panicle dry weight per plot, GWDW/plot-grain dry weight per plot, PHI-Panicle harvest index, GNP/plot- grain number per plot, GNPP-Grain number per panicle. Line numbers: 304-305
Line 508: regionse change to regions
Author’s response: Edited. Line numbers: 530
Line 652: to identify – delete, repetition
Author’s response: Edited. Line numbers: 758
References:
Line 783: Sorghum change to Sorghum
Author’s response: Edited. Line numbers: 848
Line 797: Oryza sativa change to Oryza sativa
Author’s response: Edited. Line numbers: 862
Line 803: Sorghum bicolor change to Sorghum bicolor
Author’s response: Edited. Line numbers: 868
Line 806: J plant change to J Plant
Author’s response: Edited. Line numbers: 871
Line 821: Sorghum bicolor change to Sorghum bicolor
Author’s response: Edited. Line numbers: 886
Line 827: Sorghum bicolor change to Sorghum bicolor
Author’s response: Edited. Line numbers: 892
Line 835: Sorghum bicolor change to Sorghum bicolor
Author’s response: Edited. Line numbers: 900
Line 839: Sorghum bicolor change to Sorghum bicolor
Author’s response: Edited. Line numbers: 904
Line 854: Sorghum bicolor change to Sorghum bicolor
Author’s response: Edited. Line numbers: 918
Line 860: Drosophila melanogaster change to Drosophila melanogaster
Author’s response: Edited. Line numbers: 924
Line 883: Sorghum bicolor change to Sorghum bicolor
Author’s response: Edited. Line numbers: 947
Round 2
Reviewer 1 Report
.